# SAKE: Strobemer-assisted *k*-mer extraction

**Miika Leinonen**📧*, **Leena Salmela***

Department of Computer Science, University of Helsinki, Helsinki, Finland

* miika.leinonen@helsinki.fi (ML); leena.salmela@helsinki.fi (LS)

## Abstract

*K*-mer-based analysis plays an important role in many bioinformatics applications, such as *de novo* assembly, sequencing error correction, and genotyping. To take full advantage of such methods, the *k*-mer content of a read set must be captured as accurately as possible. Often the use of long *k*-mers is preferred because they can be uniquely associated with a specific genomic region. Unfortunately, it is not possible to reliably extract long *k*-mers in high error rate reads with standard exact *k*-mer counting methods. We propose SAKE, a method to extract long *k*-mers from high error rate reads by utilizing strobemers and consensus *k*-mer generation through partial order alignment. Our experiments show that on simulated data with up to 6% error rate, SAKE can extract 97-mers with over 90% recall. Conversely, the recall of DSK, an exact *k*-mer counter, drops to less than 20%. Furthermore, the precision of SAKE remains similar to DSK. On real bacterial data, SAKE retrieves 97-mers with a recall of over 90% and slightly lower precision than DSK, while the recall of DSK already drops to 50%. We show that SAKE can extract more *k*-mers from uncorrected high error rate reads compared to exact *k*-mer counting. However, exact *k*-mer counters run on corrected reads can extract slightly more *k*-mers than SAKE run on uncorrected reads.

## Introduction

*K*-mers play an important role in bioinformatics. For example, *k*-mers are used in alignment-free sequence analysis and applications relying on de Bruijn graphs. In sequence analysis, the *k*-mer spectrum of a read set can be used for genetic variant calling [1], metagenomic classification [2], and repeat analysis [3, 4]. On the other hand, genome assembly and read correction are some of the applications that take advantage of de Bruijn graphs.

The outcome of these applications is greatly affected by the length of the *k*-mers. If *k* is small, we will usually find fewer unique *k*-mers. With fewer unique *k*-mers, it can be difficult to make meaningful inferences from the *k*-mer spectrum. Short *k*-mers also make smaller de Bruijn graphs (fewer nodes and edges). While the graph is easier to handle when it requires less space, it is harder to reconstruct the underlying sequence correctly because the graph will contain many branching paths. On the other hand, with longer *k*-mers a *k*-mer spectra can give us more meaningful information, and it will also be easier to find longer non-branching paths in de Bruijn graphs. In practice, the ability of current methods to extract long *k*-mers is limited by the sequencing error rate, and thus many tools take advantage of short *k*-mers only.

**Data Availability Statement:** Proposed method SAKE is available on GitHub (https://github.com/ Denopia/SAKE). E. coli read set used in the experiments is available on figshare (https:// figshare.com/articles/dataset/Ecoli_K12_MG1655_ R10_3_HAC/11823087). E. coli reference genome

GCA_000005845.2 used in the experiments is available from NCBI (https://www.ncbi.nlm.nih.gov/assembly/GCF_000005845.2/). D. melanogaster reference genome GCA_000001215.4 used in the experiments is available from NCBI (https://www.ncbi.nlm.nih.gov/assembly/GCF_000001215.4/). Human chromosome 1 read data used in the experiments is available from EPI2ME Labs (https://labs.epi2me.io/gm24385_2021.05/). Human chromosome 1 reference CM039011 used in the experiments is available from NCBI (https://www.ncbi.nlm.nih.gov/nuccore/CM039011).

**Funding:** This work is supported by the Academy of Finland (https://www.aka.fi/en/), via grant 323233 (LS). Open access funded by Helsinki University Library. Academy of Finland had no role in study design, data collection and analysis, decision to publish, or preparation of the manuscript.

**Competing interests:** The authors have declared that no competing interests exist.

The optimal *k*-mer length for each application is not trivial to estimate. For some sequence analysis tasks, it can be enough to use *k*-mers of length 32bp or less (this is often a convenient restriction for programs since we can encode a 32-mer in a single 64-bit integer using two bits for each character). However, for example in genome assembly, longer *k*-mers can be very advantageous [5, 6].

It is known that a genome tends to contain repeated sequences. Very short repeats are bound to happen by accident due to the alphabet size of four, but a genome also contains longer repeats. Repeats can be divided into two categories: local and global repeats. A local repeat appears multiple times in succession, and a global repeat has multiple copies in different parts of the genome. In de Bruijn graph-based genome assembly, local repeats may cause a cycle if the repeat is longer or as long as the *k*-mer length. In this case, we know what characters the underlying sequence contains, but we are unsure how many times the repeating part occurs. With global repeats, some nodes in the graph are shared between different sections of the genome. This causes ambiguous branching and we cannot be sure which paths in the graph should be used to reconstruct the underlying sequence.

To overcome the problems emerging from repeated sequences, we can increase the length of the used *k*-mers. If a section with a local repeat is shorter than the *k*-mer, the de Bruijn graph does not form a loop. On the other hand, if a globally repeating sequence is shorter than *k*, it is captured by the *k*-mers with information about its surroundings, and the repeated sequences from different genomic regions do not share nodes in the graph. Unfortunately, some repeats in the genome can be thousands of base pairs long, and at least at the moment, it does not seem feasible to utilize *k*-mers of that length. However, there are also repeated sequences that are hundreds or tens of base pairs long, and utilizing *k*-mers that contain these kinds of repeats with some context around them could be possible. Therefore, especially with genome assembly in mind, it seems beneficial to use *k*-mers that are as long as possible.

To obtain the *k*-mers of a read set, we can simply look at the reads and report all distinct *k*-mers and how many times each of them appears. Due to the many *k*-mer use cases, many approaches have already been developed to do this task [7]. However, when the reads contain sequencing errors, the length of the *k*-mers that can be extracted by these approaches is ultimately limited by the sequencing error rate.

Two approaches [8, 9] based on spaced seeds [10, 11] have been proposed to overcome substitution errors. Zentgraf and Rahmann [9] extract gapped *k*-mers instead of continuous *k*-mers, while our previous method, LoMeX [8], extracts long *k*-mers in the presence of substitution errors. LoMeX uses spaced seeds to identify and group similar *k*-length sequences in the reads and uses a simple method to call a consensus of these sequences, which is then reported as a consensus *k*-mer. The idea behind this approach is that by grouping similar sequences we assume they originate from the same genomic sequence and all the small differences between them come from sequencing errors. Due to the use of spaced seeds, LoMeX is limited to handling substitution errors and thus is not suited to process data with insertions and deletions. Because insertions and deletions are common error types in long read sequencing technologies such as Oxford Nanopore sequencing, this limitation is an important concern in practice.

In this work, we propose SAKE (Strobemer-Assisted K-mer Extraction), a method to extract long *k*-mers from reads with insertion, deletion, and substitution errors. The key techniques in SAKE use a variant of strobemers [12] to identify and group sequences of common genomic origin in a set of reads and partial order alignment (POA) [13, 14] to compute the consensus *k*-mers of the sequences.

Strobemers are an alternative to spaced seeds, which is a well-known technique in sequence analysis to deal with substitution errors. A spaced seed is a pattern that instructs us to only consider characters at fixed positions of a sequence. Thus, spaced seeds do not work well in

the presence of insertion and deletion errors as they cause the characters in a sequence to shift positions. Like a spaced seed, a strobemer is also determined by patterns appearing in the text. However, in strobemers, the patterns are found within certain windows of the text, but not at fixed positions like in spaced seeds. Therefore, strobemers are quite robust against slight shifts of positions caused by insertions and deletions.

The consensus-building phase of SAKE is similar to the correction of sequencing errors in reads and thus similar approaches are applicable here. When building the consensus sequences, we rely on the well-known multiple sequence alignment technique called partial order alignment (POA) [13, 14]. POA represents the sequences and their alignments as a graph and the consensus sequence can then be read from a path in the graph. POA has previously been applied for example in sequencing error correction [15] and in genome assembly to compute the consensus sequence [16, 17].

Our experiments on simulated data show that SAKE can extract *k*-mers accurately from read sets up to an error rate of 6%. In these data sets, the best recall of SAKE with the used parameter values is over 90%, whereas the recall of DSK [18], a conventional *k*-mer counter relying on exact occurrences of *k*-mers, drops to 20% or below with long *k*-mers. The recall of LoMeX behaves similarly to that of DSK. On the other hand, the precision of SAKE and DSK remains similar, while LoMeX precision is the lowest. On a real *E. coli* data set with an error rate of 7.8%, the recall of SAKE remains over 90% with the best parameter values but the precision drops to 74% on 97-mers. On the other hand, the recall of DSK on 97-mers on this data set drops to 52% or lower, though its precision is higher than SAKE. LoMeX has a recall rate similar to DSK, but again its precision is the lowest of the three methods. We found out that using an error correction algorithm on the reads before giving them to DSK gave slightly better results than just using SAKE, but correcting full reads can be an unnecessarily expensive operation timewise since only the *k*-mers are required to be correct.

## Definitions and problem statement

Before we can apply *k*-mer-based methods, we need to find which *k*-mers are present in the genome. Often this is done with exact *k*-mer counting.

**Definition 1 (*k*-mer counting)** *Given a set of reads R, find all unique k-mers occurring in the read set and for each k-mer report the number of occurrences in the read set.*

The ultimate goal is to find all the *k*-mers that occur in the genome. Due to sequencing errors, not all found *k*-mers are correct. By also counting how many times the *k*-mers appear in the read set, we can estimate which of them are likely to also appear in the genome. We define a *k*-mer minimum abundance threshold *a*. If a *k*-mer appears at least *a* times, we can assume it is also present in the genome since it is strongly supported by the data. Many applications based on *k*-mer spectra can benefit from reported *k*-mer counts. On the other hand, in de Bruijn graph methods we are not that interested in the specific *k*-mer counts as long as we can trust the reported *k*-mers are part of the genome. We call the search for these *k*-mers *k*-mer extraction.

**Definition 2 (*k*-mer extraction)** *Given a set of reads sequenced from a genome, deduce the set of k-mers present in the genome.*

There is only one correct answer for *k*-mer extraction, but it is very difficult to find in practice because the data is never fully accurate. The presented *k*-mer extraction definition does not require the counts to be reported, but *k*-mer counting can be used to solve the problem. The *k*-mer extraction definition could be extended to also report an estimated number of times each *k*-mer appears in the genome. This way it would also be more suitable for *k*-mer spectrum applications. However, in this paper, we focus on the simpler *k*-mer extraction

problem definition, which still gives useful results for applications like the ones utilizing de Bruijn graphs.

*k*-mer counting would be a perfectly fine solution to *k*-mer extraction if the given data was error-free. However, the machines we use to read biological sequences are not completely accurate. For example in genome sequencing, while it is possible to get relatively low error rates with short reads, the machines that produce longer reads tend to have increased error rates. Using a minimum threshold *a* with exact *k*-mer counting can work when we are searching for shorter *k*-mers (standard 32bp and shorter) since it is likely that with enough read coverage we can find enough error-free occurrences of the correct *k*-mers. But once we raise the value of *k*, *k*-mer counting becomes unreliable. The longer the *k*-mers are, the more likely it is that they do not appear without errors in the reads.

The first solution one might think of is to first correct the errors in the data and then use the exact *k*-mer counting method. This is definitely an easy choice that will improve the results. However, correcting full reads is a task that can take a lot of time. The question we aim to answer in this work is if it is possible to extract *k*-mers and perform the error correction only on the *k*-mers during the *k*-mer searching process. Our intuition is that this kind of approach would be more efficient and lightweight than unnecessarily correcting the full reads before doing exact *k*-counting. Therefore, we aim to find out if our proposed approach meaningfully improves the precision and accuracy of the reported *k*-mers when we are extracting *k*-mers from reads that contain substitution, insertion, and deletion errors.

## Related work

SAKE uses a similar pipeline as LoMeX [8]: First we identify and group common origin *k*-mers and then we find the consensus between the grouped *k*-mers. However, LoMeX is limited to reads with only substitution errors, whereas SAKE allows insertions and deletions too. To do this, we devise an improved way to find common origin *k*-mers, and to find their consensus. The *k*-mer grouping step is done with strobemers [12] instead of spaced seeds, and the consensus *k*-mer step is done using partial order alignment [13] instead of position-wise character counts. In this section, we give an overview of LoMeX [8] and review the building blocks of SAKE: strobemers and partial order alignment.

**LoMeX.** The development of SAKE was motivated by our interest in improving the LoMeX [8] method. LoMeX is a *k*-mer extraction program that can find long *k*-mers in erroneous sequencing data more accurately than the standard counting method. LoMeX is based on finding consensus *k*-mers between *k*-mers in the reads that are believed to originate from the same genomic *k*-mer. The *k*-mers in the reads are grouped using spaced seeds [10, 11]. If two *k*-mers have the same characters in the fixed positions indicated by the spaced seed, LoMeX assumes they correspond to the same genomic *k*-mer. *k*-mers with the same origin can be grouped correctly even if the *k*-mers have substitution errors in the "don't care" positions of the spaced seed.

Next, *k*-mers in the same group are "stacked" on top of each other. Then, column by column, LoMeX finds the consensus between the *k*-mers to determine which characters should be used for the consensus *k*-mer positions. After all *k*-mer groups have been processed this way, the produced consensus *k*-mers are reported as the output of the program.

We found out that LoMeX is capable of finding long *k*-mers more accurately than DSK [18] which uses the exact *k*-mer counting method. In these experiments, the prevalent error type in the reads was substitution, which is the intended use case for LoMeX. The drawback of LoMeX is that it is not well suited for data where the errors also contain a non-insignificant number of insertions and deletions. The reason for this is that if a *k*-mer in a read contains

such errors, the characters that are intended to be at fixed positions might move to a "don't care" position and vice versa. Thus, it becomes unlikely that we can correctly group *k*-mers of the same origin through spaced seeds.

**Strobemers.**   To group similar sequences in the presence of insertions and deletions, we cannot use spaced seeds at fixed positions. Another popular approach to find candidate sequences for alignments is to use minimizers. A minimizer is the smallest *k*-mer in a specific text window. To determine which *k*-mer is the smallest one, any ordering can be used. For example, using lexicographical order to find the 3-mer minimizer in text **TAGAACC** would mean that the minimizer is **AAC**. SAKE utilizes strobemers to group similar sequences. A strobemer [12] can be seen as a spaced seed where the fixed positions characters are replaced by minimizers. This makes it possible to identify matching sequences even if they contain some insertions and deletions.

Specifically, a strobemer $S_i$ at position *i* in a sequence *s* is a concatenation of *n* sequences $m_1, m_2, \ldots, m_n$ (called strobes), each of length *l*. The first strobe $m_1$ is the *l*-mer starting at position *i*, i.e. $m_1 = s[i: i + l - 1]$. All the following strobes $m_2, m_3, \ldots, m_n$ are minimizers that come from windows defined by parameters $w_{min}$ and $w_{max}$. For all $m_i$ with $i > 1$, $m_i$ is the minimizer between all *l*-mers whose first character is found in window $s[i + w_{min} + (n - 2) w_{max}: i + w_{min} + (n - 1)w_{max}]$. The full strobemer $S_i$ is simply a concatenation of the first *l*-mer $m_1$ and the minimizers $m_2, m_3, \ldots, m_n$ in order. Fig 1 (middle) illustrates the definition.

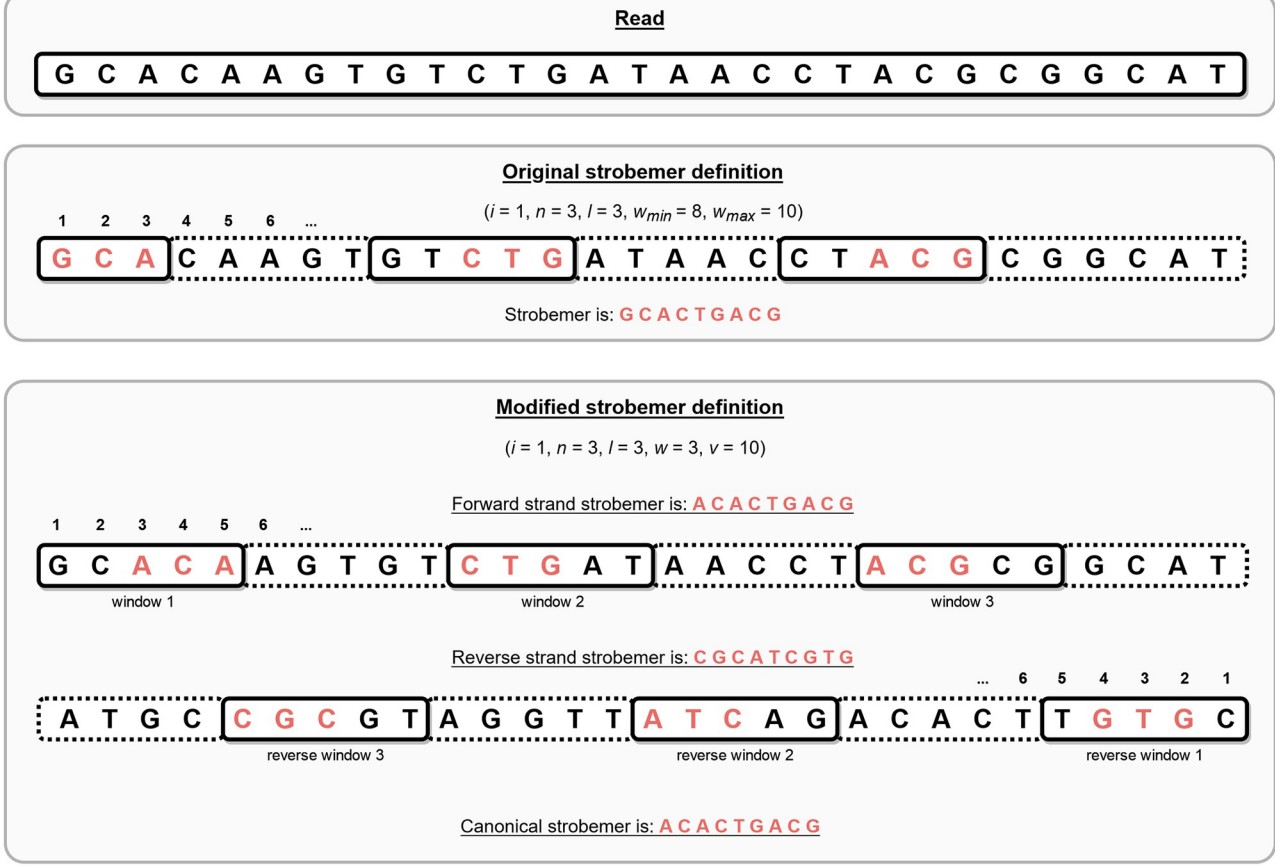

**Fig 1. Strobemer definition example.** An example read (top), and how to find the strobemer at position *i* = 1 according to the original strobemer definition (middle) and our modified strobemer definition (bottom).

Due to the challenges emerging from reverse complement sequences, we did not use the original strobemer definition directly. Instead, we modified the definition slightly to better suit SAKE. We will explain the differences between the original definition and our modified one in the Methods section.

**Partial order multiple sequence alignment.** As our method needs to accommodate also insertions and deletions, we cannot use the same consensus *k*-mer generation method as in LoMeX. Thus, we use partial order alignment (POA) [13] for the multiple sequence alignment (MSA) and consensus sequence generation step. POA is used to build an MSA graph for given sequences, and then a consensus sequence generation algorithm [14] finds the consensus between the sequences.

Partial order multiple sequence alignment (PO-MSA) graph generation works by aligning and adding sequences to the graph one by one. First, the current sequence is transformed into a simple one-dimensional graph where each character is a node, and consecutive character nodes are connected with a directed edge. Next, this one-sequence graph is aligned to the PO-MSA graph using a dynamic programming solution. The sequence-to-graph alignment algorithm works similarly to a sequence-to-sequence alignment algorithm, except in a graph a character attached to a node can have multiple "previous characters" (nodes corresponding to incoming edges). So, when the optimal score for each cell in the dynamic programming matrix is calculated, there can be more than the usual three options (insertion, deletion, match).

Once the optimal alignment between the current sequence and the PO-MSA graph is found, the graph of the sequence is added to the PO-MSA graph. This happens by fusing aligned nodes and then deleting unnecessary edges. If a node in the sequence is aligned to a node in the PO-MSA graph and these nodes have the same character, the nodes are fused. If the aligned sequence and PO-MSA nodes have different characters, but the PO-MSA node is already aligned with another PO-MSA node that has the same character as the sequence node, then those nodes are fused instead. Lastly, if the sequence and PO-MSA nodes are aligned and have different characters, and the PO-MSA graph node is not aligned to a node whose character matches the sequence node character, no nodes are fused. Instead, they are marked as aligned nodes. If a node in the sequence is not aligned to any node in the PO-MSA graph, then that node simply becomes a new unaligned node in the graph. After the graph fusion, if nodes that have more than one edge between them are found, the excess edges are removed.

During the graph construction process, information about the sequence ID and position in the sequence is stored in each node. This way it is possible to traverse the graph and reconstruct any sequence used to build it, so no information is lost.

**Consensus sequence generation.** After the PO-MSA graph is ready, it can be used to generate consensus sequences. For example, Lee [14] introduces one such algorithm called *heaviest bundle*. The algorithm works by processing nodes in the graph in topological order and calculating optimal scores for them. The weight of an edge is defined as the number of sequences in the PO-MSA graph which contain this edge. First, the starting node, i.e. node with no incoming edges, is given a score of zero. When the algorithm assigns scores to the rest of the nodes, it looks at all the incoming edges and picks the one with the highest weight. Then, the algorithm assigns a score to the node that is the sum of the weight of the heaviest incoming edge and the score of the node that the edge is coming from. After all node scores have been calculated, the algorithm picks the node with the highest score and traces back the sequence of nodes contributing to its score. The sequence spelled by these nodes is the consensus sequence.

The next step is to bundle sequences with the consensus sequence. This means that the algorithm looks at all sequences in the graph and determines if they are similar enough to the consensus sequence to be bundled with it. A sequence of length $\ell$ is deemed similar enough if its

path through the PO-MSA graph shares at least $b \cdot \ell$ nodes with the consensus sequence path, where $b$ is the bundling threshold parameter (with value between 0 and 1). The sequences that are bundled are the ones that most likely contributed to the generation of the consensus sequence. A more detailed explanation of the bundling step can be found in the original publication [14].

Finally, the weight contributions of the bundled sequences in the graph are adjusted. For example, if we do not want to use them to generate more consensus sequences, the algorithm can subtract 1 from the edge weights for each bundled sequence that uses that edge. Essentially, this completely removes the bundled sequences from the graph. After bundling and edge reweighting, the process starts from the beginning. The algorithm stops after no sequences are bundled to the current consensus sequence because otherwise no reweighting would happen and the same consensus sequence would be found again in the next round. In the end, the algorithm reports all found consensus sequences, their bundled sequences, and the MSAs of each bundle.

## Methods

In the Introduction section, we introduced the two key concepts behind SAKE: strobemers [12] and POA [13]. Now, we will present our contribution by giving a more detailed description of SAKE.

### Overview of SAKE

Counting exact *k*-mers in a read set with a high error rate is unlikely to yield accurate results especially when the value of *k* is high. Still, with a high enough coverage, the read set should contain a decent number of copies of all genomic *k*-mers, even if some instances contain few errors. To find what these genomic *k*-mers are without the errors introduced by imperfect sequencing, SAKE tries to find all *k*-mers in the reads that correspond to the same genomic *k*-mer and use their consensus to infer what the error-free genomic *k*-mer looks like.

Our solution to the *k*-mer extraction problem has two main phases: (i) identify and group sequences in the reads that correspond to the same *k*-mer in the genome and (ii) find the consensus between the sequences to solve what the correct *k*-mer is. In LoMeX, spaced seeds were used to identify a feature in a *k*-mer (called spaced *k*-mer), and *k*-mers that shared the same feature belonged to the same group. Spaced seeds work when the only type of error in the data is substitution. If a notable amount of insertions and deletions are also present, grouping read *k*-mers correctly becomes difficult. Additionally, due to these error types, the sequences in the reads that correspond to a genomic *k*-mer are not necessarily of length *k*. For these reasons, we need to use some other feature than spaced seeds that allows us to group sequences with a common origin, even in the presence of insertions and deletions.

As mentioned, in LoMeX we used spaced *k*-length patterns with certain fixed positions as features to identify and group *k*-mers with a common origin. The *k*-length pattern, or spaced seed [10, 11], is applied to all *k*-mers in the reads, and the *k*-mers with matching characters in the fixed positions are put in the same group. For example, if we have 5-mers ATTGC and ACTCC and the spaced seed is 10101 (where ones mark the fixed positions), then both of these 5-mers yield the same spaced 5-mer A*T*C and they are put into the same group. This method works well when the most prevalent error type is a substitution, but when a considerable number of insertions and deletions are also present, it becomes more unlikely to group *k*-

mers correctly. Even in the non-fixed positions, insertions and deletions can shift the characters in fixed positions, causing us to group *k*-mers incorrectly.

To overcome the problem of insertions and deletions, we take a different approach where the use of spaced seeds is replaced by strobemers [12]. However, our definition of strobemers slightly differs from the original definition introduced in the Introduction section to make it more suitable for SAKE. Nevertheless, we will use the term strobemer to refer to our modified definition, which will be introduced in the following Strobemer search section.

SAKE pipeline starts by finding all strobemers that appear in the input reads. Once all solid strobemers (that appear at least *a* times where *a* is a threshold set by the user) are found, they are used to group sequences that are likely from the same genomic origin. The lengths of the grouped sequences would ideally be *k*, but in reality, the lengths vary because of insertions and deletions. Therefore, we cannot simply stack the sequences on top of each other and solve the consensus by counting the most common character column by column like it was done in LoMeX. Instead, we have to solve the multiple sequence alignment and consensus sequence generation in a different way.

To find the consensus *k*-mers we align the grouped sequences using partial order alignment (POA) [13] and then generate a consensus sequence from the graph. We use the POA method to build a partial order multiple sequence alignment (PO-MSA) graph using a ready implementation called SPOA [19]. The heaviest bundle algorithm provided by the implementation to generate the consensus sequence did not produce good enough sequences for our method so we devised a new more suitable one. We will explain how we used POA and what modifications we made to the consensus generation algorithm in more detail in the Consensus sequence generation section.

One drawback in using strobemers is that the consensus sequences are not always of length *k* because the input sequence lengths vary. For this reason, if a consensus sequence is longer than *k*, we have to output all *k*-mers that appear in it. This means that sometimes two different consensus sequences from different groups can produce the same *k*-mer, so we need a cleanup step at the end where duplicate *k*-mers are removed from the output.

There is also the opposite concern: what if a consensus sequence is shorter than *k*? In this case, SAKE would be unable to produce any consensus *k*-mers from that sequence. To avoid this problem, we make sure that all sequences used in consensus generation are at least of length *k*. Thus, it should be very unlikely that a consensus sequence shorter than *k* is produced, which was also supported by our observed consensus sequences. The length of the desired *k*-mers affects how long the input sequences need to be for the consensus generation, which is then determined by the strobemer parameters. We will discuss how to choose the strobemer parameters in the Output *k*-mer splitting section so that the strobemers find long enough sequences for the consensus generation.

In summary, our proposed method SAKE contains the following steps:

- Find all solid strobemers in the reads.

- Find where these solid strobemers appear, and group the sequences covered by the strobemers accordingly.

- Use POA to find the consensus sequences of each strobemer group.

- Split too long consensus sequences into appropriately sized *k*-mers.

- Take the canonical versions of the *k*-mers and remove duplicates.

SAKE flowchart with these steps can be seen in Fig 2.

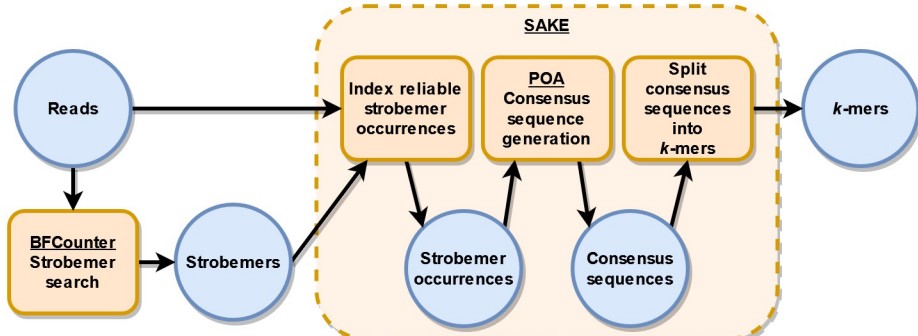

**Fig 2. SAKE flowchart.** First, we find solid strobemers from the read set with the modified BFCounter. Next, the solid strobemers are given to SAKE with the reads so it can find and index the sequences covered by the solid strobemers. Then, with the help of POA, consensus sequences are generated for each strobemer using the sequences they cover in the reads. Finally, the consensus sequences are split into *k*-mers which are then reported to the user.

## Strobemer search

SAKE identifies and groups sequences of common origin based on their strobemers. Unfortunately, we could not use the original strobemer definition. This is because we needed a strobemer definition with well-defined canonical strobemers. In other words, we want a way to extract the same strobemer from a sequence regardless of whether it is in forward or reverse complementary orientation.

In the original definition, the first part (i.e. strobe) of the strobemer is located in a fixed position, unlike the following strobes which are minimizers whose position varies. We redefined the strobemer for SAKE so that already the first strobe of the strobemer is a minimizer. Now each strobe is found from an equally long minimizer window, and the distance between consecutive minimizer windows is always the same. Therefore, the minimizer windows in the forward and reverse orientation of a sequence cover the same characters. For example, let us assume the first minimizer window in the forward orientation starts at position $i$ and looks for an $m$-length minimizer in an $s$-sized minimizer window. The last character we need to look at to determine the minimizer for this window is at position $i + s + m - 1$. On the other hand, in the reverse orientation, the last window would start at position $i + s + m - 1$ (using the forward strand indexing). The last possible minimizer in this window would start at position $i + s + m - 1 - s = i + s - 1$, and it would end at position $i$. Thus, the first minimizer window in the forward orientation and the last minimizer window in the reverse orientation use the same characters of the sequence. And because the distance between the windows in both orientations is the same, the second window in the forward orientation overlaps with the second to last window in the reverse orientation, and so on, until the last window in the forward orientation overlaps with the first window of the reverse orientation.

With this strobemer definition, we are considering the same characters of a sequence regardless of its orientation. This way, if the characters covered by the minimizer windows in one orientation are error-free, they are also guaranteed to be error-free in the reverse orientation. Defining a reverse strobemer that holds this property for the original strobemer definition seemed difficult since the first strobe is always at a fixed position. We decided to make this modification because it made the reverse (and canonical) strobemer definition more consistent. Consensus sequence generation also becomes more efficient if we only need to consider the canonical strobemer instead of two strobemers from both orientations.

Another reason we wanted to change the definition was that because in the original one the first strobe comes from a fixed position, it is unlikely that strobemers from consecutive positions are identical. Conversely, in the modified definition all the strobes are minimizers, which means that we can expect to sometimes see identical strobemers from consecutive read positions. This leads to fewer different strobemers being extracted, making the consensus generation even more efficient.

Because of the strobemer definition modification, we also had to redefine its parameters. In our definition, the parameter *w* expresses the length of the minimizer window. By the length of the window, we mean how many minimizer candidate starting positions the window has (not the number of minimizer candidates that fit in the window). Parameter *v* shows the distance between two consecutive minimizer window starting positions. And finally, we have the parameters *n* and *l* also present in the original definition to show how many minimizers are needed for the strobemer concatenation and how long the minimizers should be, respectively. Here is the exact definition of the modified strobemer:

**Definition 3 (Strobemer)** *A strobemer $S_i$ in string s at position i with parameters n, l, v, and w is a concatenation of n sequences $m_1$, $m_2$, . . ., $m_n$, where sequence $m_j$ $1 \leq j \leq n$ is the minimizer of all l-length sequences with a starting position in window $s[i + (j − 1)v: i + (j − 1)v + w − 1]$.*

In Fig 1 we can see an example of a read, and how we can find a strobemer according to the original definition (middle) and our modified definition (bottom) with some arbitrarily chosen parameters.

To find a strobemer, we need to build it using minimizers. Defining a minimizer requires a metric to determine the order of the sequences. The simplest way is to use alphabetical order. This is not the optimal choice though, since alphabetical order seems to favor specific minimizer window positions and it can also lead to some unintended behavior with real data due to sequencing errors in homopolymer regions. In the original strobemer publication [12], different versions of the strobemer were defined based on how the minimizer was determined. Here we use the *minstrobe* approach, where minimizers between different windows are independent of each other. We use an invertible hash function (https://naml.us/post/inverse-of-a-hash-function/) to determine the minimizer which is already used e.g. in minimap [20].

Before we can group sequences in the reads that have a common genomic origin based on strobemers, we first need to determine which strobemers exist in the read set and which ones are solid. This step is very similar to the classic *k*-mer counting process: find all strobemers and their counts, and only accept strobemers where the count is at least equal to a threshold value *a*. These strobemers are solid. We chose to use BFCounter [21] to count the occurrences of strobemers in the reads because it can handle larger *k*-mers and it was relatively easy for us to modify the code to search strobemers instead of regular *k*-mers. BFCounter is a *k*-mer counting tool that utilizes a Bloom filter to first find *k*-mers that appear at least twice and then count how many times exactly those *k*-mers are found in the reads. After our modification, the program uses the same counting method but finds strobemers instead of *k*-mers. Instead of inputting the desired *k*-mer length, the modified BFCounter must be given the strobemer parameters (*n*, *l*, *w*, *v*). As an output, it produces all strobemers that appear at least *a* times and the exact number of times they occur.

## Strobemer occurrence gathering

After the modified BFCounter has found all solid strobemers, they are given to SAKE. The user can set the value for abundance threshold *a*, but it should be at least 2 to filter out erroneous strobemers. Experimentation was made with different values of parameter *a*, and this is discussed more in the SAKE Parameter selection section.

SAKE builds a map where all solid strobemers are set as keys, and the corresponding value for each key is an empty list where the strobemer occurrences are stored. Next, SAKE starts scanning the reads to find all occurrences of the solid strobemers and adds them to the map. In this phase, SAKE looks at each position in each read. If it finds a strobemer that was reported as solid, the sequence covered by this strobemer is added to its list in the map. We define the sequence covered by a strobemer to be the part of the read that falls between the first minimizer and the last minimizer of the strobemer (including the minimizers). If multiple consecutive positions in a read produce the same strobemer, the area covered by them all is added to the map only once as a single sequence.

As an example, we can look at Fig 1 where we determined that the canonical strobemer is ACACTGACG. In this strobemer, the first minimizer is **ACA** and the last minimizer is **ACG**. The strobemer was found in the forward strand and the sequence covered by it is **ACA**AGTGTCTGATAACCT**ACG**. The reason we define the covered sequence like this is that all the sequences that are grouped under the same strobemer are identical at the beginning and end. This makes it easier to construct the PO-MSA graph for consensus generation. After this step is complete, the map holds all solid strobemers and the sequences they cover in the reads, and we are ready to move on to consensus generation.

## Consensus sequence generation

When all solid strobemers and sequences they cover are available, we can start generating consensus sequences. As mentioned in the Introduction section, we use partial order alignment (POA) [13] to solve the necessary MSAs.

First, we build a PO-MSA graph for each strobemer using the sequences the strobemer covers in the reads. The graph is built by aligning and adding sequences to the graph one sequence at a time. The nodes in the graph are labeled with characters and edges connect nodes that appear next to each other in a sequence. To implement POA we used SPOA, the fast SIMD version as implemented by Vaser et al. [19].

After all sequences are added to the graph, we can start generating consensus sequences. Unfortunately, based on our initial trials, the heaviest bundle algorithm described in the Introduction section did not produce good enough consensus sequences. We then modified the heaviest bundle algorithm so that dependencies between distant nodes in the graphs are also considered to get more accurate consensus sequences.

**Modified *heaviest bundle* algorithm.** When several regions in the genome produce the same strobemer, sequences from all of them are included in the same PO-MSA graph. To avoid consensus sequences that combine parts from different regions, we use an algorithm similar to the isoform generation mode presented by Lee [14]. The difference between our method and the isoform generation mode is the way we choose the sequences to be reweighted. Additionally, we perform a filtering step where only strongly supported consensus sequences are considered for *k*-mer generation.

We follow the same idea as in the original heaviest bundle algorithm. However, before we start calculating scores for the nodes, we aim to reweight edges that correspond to sequences from the same genomic region. Reweighting starts by finding the average sequence length between the sequences in the graph. Then, we determine a weight threshold $t_w$ for strongly supported edges. If an edge has a weight that is at least $t_w$, it is considered to be strongly supported by the data. We choose $t_w$ to be the weight of the *i*th heaviest edge among all edges in the graph, where $i$ is the average sequence length. This way, we have at least $i$ strong edges in the graph. Thus, if there is one path in the graph that is supported by most sequences, then most edges in this path should also be considered strongly supported.

After we have determined which edges in the graph are strongly supported, we can find which sequence in the graph is most strongly supported. To do this, we calculate a weakness score $s_w$ for each sequence in the graph. The weakness score should be low when the path of the sequence in the graph contains a lot of strong edges. On the other hand, the weakness score should be high when the sequence contains a lot of edges that are not strongly supported. Combining these two aspects makes it so we do not discriminate against shorter sequences. Longer sequences are naturally more likely to contain many strong edges, but their weakness score is increased for each weak edge too. The weakness score for a sequence is defined as $s_w = e_m + e_w$, where $e_m$ is the number of strongly supported edges that are not part of the sequence, and $e_w$ is the number of edges in the sequence that are not strongly supported. After weakness scores are calculated, we can find which sequence has the lowest weakness score and select that as the most strongly supported sequence.

We use the strongest sequence as a backbone for consensus generation. We do not want to simply raise the weight of the edges that correspond to the strongest sequence because that would make the process too biased towards one specific sequence (and any errors within it). Instead, we find other sequences that are similar to the strongest sequence. These sequences are likely from the same genomic region, and if we give bonus weight to them all, we can favor a specific genomic region without being biased toward one specific sequence. In order to find these similar sequences, we first raise the weight of the edges of the strongest sequence. Then, we use the consensus generation algorithm to find an initial consensus sequence and its associated bundle of sequences. In this step, the consensus sequence is likely to be identical to the strongest sequence since that was the only one given bonus weight. The generated bundle contains sequences that are similar to the strongest sequence, and we raise the weights of these bundled sequence edges. After reweighting the edges, we are now focusing on a group of sequences that are likely to originate from the same genomic region.

Next, we can move on to generating the actual consensus sequence. Again, this is done using the original SPOA implementation of this step. Once we find the consensus sequence and its bundled sequences, they are recorded to be reported later. All the edges whose weight was raised before are returned to normal, and sequences that are part of the consensus sequence bundle are removed from the graph by setting their weight to zero. If a proper consensus sequence is found with enough bundled sequences, we start from the beginning to generate the next consensus sequence. We do this until we cannot find a new consensus sequence.

Fig 3 shows an example POA graph and two cases of consensus generation. The top graph shows how the original consensus generation algorithm can fail by finding an incorrect consensus sequence. In the bottom graph, the modified algorithm has given additional weight to a set of sequences that originate from the same genomic region, and we are able to find the correct consensus sequence.

**Consensus sequence filtering.**   We do not accept all consensus sequences produced from a graph. To filter out bad consensus sequences, we take a look at their bundled sequences. During the consensus sequence generation, the heaviest bundle algorithm considers all the sequences in the graph. Using the default or user-specified bundling threshold $b$, a sequence with length $\ell$ is added to the bundle if it shares at least $b \cdot \ell$ nodes with the consensus sequence. The purpose of the bundled sequences is to support the existence of the consensus sequence in the genome. In order to verify that a consensus sequence has enough support, the algorithm first requires that the bundle contains enough sequences. There are two different thresholds for the bundling support: absolute threshold $g$ and proportional threshold $z$. Let us assume we have a consensus sequence $C$, and $B$ is the set of sequences in its bundle. Now, $C$ is considered to have large enough bundle (and thus support from the input data) if $|B| \geq max(g, z|S|)$, where $S$ is the set of all sequences corresponding to this specific strobemer.

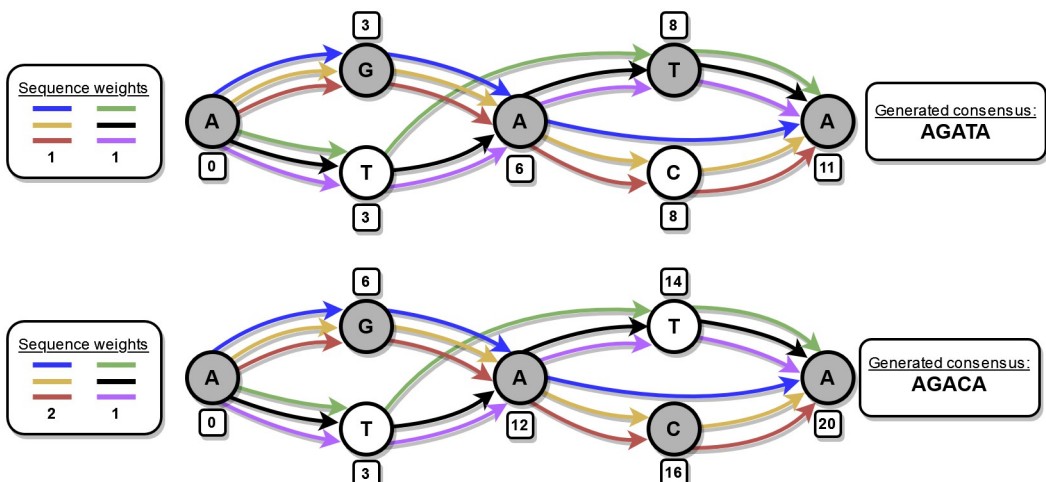

**Fig 3. A very simple POA graph example.** The graph represents two sequences from different positions of the genome: AGACA and ATATA. The graph consists of six sequences that originate from these two genomic sequences: AGAA (blue), AGACA (yellow), AGACA (red), ATTA (green), ATATA (black), and ATATA (purple). Blue and green sequences have an error in them, specifically one deletion in each making them skip a node. The numbers next to the nodes show the node weight calculated according to the heaviest bundle algorithm. Note that in a real POA graph there can be only one edge between two nodes, but in this example, we have drawn edges for each individual sequence to make the example clearer. You can think of the weight of an edge between two nodes in this picture to be the number of edges between the nodes. If we try to find the consensus sequence using the original heaviest bundle algorithm (top) so that all sequences have the same weight, we will generate an incorrect sequence AGATA which is a combination of the two genomic sequences represented in the graph. On the other hand, if we use our modified version (bottom), we can find a correct consensus sequence AGACA if we weight the sequences properly. The algorithm detects that blue, yellow, and red sequences are similar and gives them additional weight. This way the path corresponding to their genomic origin sequence is favored and we can generate it successfully.

In addition to the bundle size filtering, we also require each character in the consensus sequence to have enough support from the sequences in its bundle. For this, we look at the MSA of a consensus sequence and its bundle sequences that is provided by the SPOA implementation. For each position in the consensus sequence, we check if the character appears at least $c$ times in the bundled sequences as well. Dashes (deletions) that appear in the MSA for the consensus sequence are ignored. This just means that some bundled sequences have insertions that do not appear in the consensus sequence. If all the characters in the consensus sequence are supported by at least $c$ bundled sequences, it is accepted as a consensus sequence.

Using this modified version of the heaviest bundle algorithm with additional filtering gives us a set of consensus sequences that we would like to report as the found *k*-mers. The problem is, that not all consensus sequences are exactly *k* characters long. We still need to add one step where consensus sequences are transformed into *k*-mers.

## Output *k*-mer splitting

Because the consensus sequences are not necessarily of length *k*, we need to cut them into appropriately sized *k*-mers. The strobemer parameters $n$, $l$, $v$, and $w$ are chosen so that sequences used in PO-MSA are at least *k* characters long. Thus, it is unlikely that a consensus sequence would be shorter than *k*. During our experimentation, we observed that input sequences that were at least *k* characters long produced hardly any consensus sequences shorter than *k*. For example, with a simulated 40x 4% error rate read set, a common occurrence was that only about 0.001% of the consensus sequences ended up being too short.

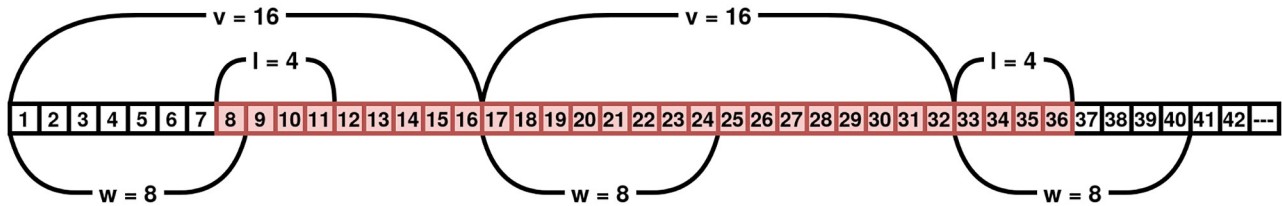

**Fig 4. Strobemer parameters and how they affect the minimum area covered by the strobemer.** Squares represent characters and the number in the square is the index of the character. With parameters $n = 3$, $l = 4$, $w = 8$, and $v = 16$ the shortest area the strobemer at position $i = 1$ can cover is $(3 - 1) \times 16 - 8 + 4 + 1 = 29$ characters long. The colored squares represent the shortest sequence the strobemer can cover.

We also want the consensus sequences to be close to the desired length so the need for cutting them into *k*-mers would be minimized. If the consensus sequence is too short, which is a very unlikely case, it cannot be used to produce a *k*-mer. If the length of the consensus sequence is exactly *k*, we can just accept it as it is. In the case where the consensus sequence is too long, we simply take all *k*-mers that appear in it. It is usually the case that *k*-mer counters return only the canonical *k*-mers because the strand is unknown. We also do this after consensus sequences are split into *k*-mers. Because multiple sequences can produce the same *k*-mer we will only report one such *k*-mer in its canonical form.

To guarantee that input sequences for PO-MSA are of length at least *k*, we need to set strobemer parameters properly. As we explained earlier in the Strobemer occurrence gathering section, the sequence covered by a strobemer starts where the first minimizer begins, and ends where the last minimizer ends. To decide what strobemer parameters to use, we need to know what is the shortest sequence a strobemer can cover. Given strobemer parameters $n$, $l$, $v$, and $w$, the shortest possible sequence the strobemer can cover has length $(n - 1) \times v - w + l + 1$. This happens when the first minimizer is at the last position of its window and the last minimizer is at the first position of its window. For example, in Fig 4 with parameters $n = 3$, $l = 4$, $v = 16$, and $w = 8$, the smallest possible area covered is $(3 - 1) \times 16 - 8 + 4 + 1 = 29$ characters long. So, if we were searching for 29-mers, these parameters would be suitable since it is unlikely we will produce consensus sequences shorter than 29. In the general case, we can define the following inequality relationship between *k* and the strobemer parameters:

$$(n - 1) \times v - w + l + 1 \geq k \tag{1}$$

For any given *k*, it is desirable to choose the strobemer parameters so that they just barely satisfy the inequality. This way, the generated consensus sequences are more likely to be close to *k* in length to avoid unnecessary work producing duplicate consensus *k*-mers. We will discuss how the SAKE parameter values were chosen in the following SAKE parameter selection section.

## SAKE parameter selection

SAKE has a few parameters the user can adjust to affect the precision and recall of the reported *k*-mers. Also, the modified BFCounter has new parameters to determine the shape of the strobemers. These parameters are listed in Table 1. Before the experiments shown in the Results section were performed, we needed to find what values to use for these parameters.

Our goal was not to fully optimize the parameters since this would have taken unnecessarily long. Instead, we were satisfied with using a reasonable set of parameter values if tweaking them would not significantly improve the precision and recall of the produced *k*-mers. For

**Table 1. SAKE and modified BFCounter parameters and their descriptions.**

| Parameter | Program | Type | Description |
|---|---|---|---|
| $n$ | BFCounter | Positive integer | Number of minimizers used to determine a strobemer. |
| $l$ | BFCounter | Positive integer | Strobemer minimizer length. This is the length of the strobes that are used to construct a strobemer. |
| $v$ | BFCounter | Positive integer | Distance between the beginning positions of two consecutive minimizer windows. |
| $w$ | BFCounter | Positive integer | Strobemer minimizer window size. This is the number of $l$-mers from which a minimizer, i.e. strobe, is chosen. |
| $a$ | SAKE | Positive integer | Minimum strobemer abundance. This value shows how many occurrences a strobemer needs to have in the reads for the system to start generating consensus $k$-mers for the strobemer instance. |
| $g$ | SAKE | Positive integer | Minimum bundle support threshold. This value shows the minimum number of sequences that are required to be in a consensus sequence bundle for the consensus sequence to be used in consensus $k$-mer generation. |
| $z$ | SAKE | Number between 0 and 1 | Minimum bundle support threshold relative to the bundle size. Multiplied with the number of sequences used to build a POA graph, the value shows the minimum number of sequences that are required to be in a consensus sequence bundle for the consensus sequence to be used in consensus $k$-mer generation. |
| $b$ | SAKE | Number between 0 and 1 | Bundling threshold. Used to determine if a sequence used to build a POA graph belongs to the same bundle with a generated consensus sequence. |
| $c$ | SAKE | Positive integer | Character support threshold. Used to make sure each character in a consensus $k$-mer has enough support from the bundled sequences. |

some parameters, we could reason out the values that would work well. For the rest that could not be as easily reasoned, we made initial guesses and performed experiments to see if adjusting them would make a meaningful difference. Of course, this does not necessarily lead to an optimal set of parameter values. Still, the amount of time it would have taken to find the best parameter values for a marginal increase in result quality did not seem worthwhile, especially since the "best" parameter values are dependent on the used data set. Also, it is difficult to determine what are the optimal parameter values, since there is often a tradeoff between precision and recall.

The experiments to determine parameter values were performed with multiple runs on simulated 40x 6% error rate *E. coli* reads. We reasoned out the values for parameters $n$ and $l$, but for the rest of the parameters we made the following initial guesses: $w = 11$, $a = 3$, $g = 3$, $z = 0.4$, $b = 0.8$, and $c = 3$. The value of parameter $v$ is not fixed a priori, because it was used to adjust the size of the area covered by the strobemer to fit each individual $k$-mer length according to Eq (1) in the Output $k$-mer splitting section.

First, we decided which values to use for the strobemer parameters $n$, $l$, $v$ and $w$. This choice is guided by the inequality $(n - 1) \times v - w + l + 1 \geq k$ defined in Eq (1), which ensures that the sequences that are used for consensus generation are of length at least $k$. Fixing some of the parameter values will affect what values we can choose for the other parameters. We decided that the length of the strobemer (the sum of the strobe lengths $nl$, not including the gaps between them) should not be dependent on the length of the searched $k$-mers. The reasoning behind this was that the strobemer characters are the only ones that are fixed in consensus sequence generation, and we wanted the probability that they contain an error to be the same in all experiments regardless of the $k$-mer length. Thus, we used the same value for $n$ and $l$ in all experiments.

When deciding the value for $l$, we should consider that we want it to be likely that a minimizer should be unique in its window. For parameter $n$ we also need to take into account that the number of the minimizers should be at least two (and preferably more) for the inequality (and the strobemer itself) to make sense. On the other hand, we do not want the strobemer length $nl$ to be too long, because the longer the strobemer is, the more likely it is that the strobemer contains an error. Sahlin [12] ran experiments where the total strobemer length was 30.

We tried using a similar setup with parameters $n = 3$ and $l = 10$ which leads to the same strobemer length. Unfortunately, this did not work out well enough and the recall rate was not as high as we would have hoped. We assume that in SAKE the effects of errors in the read are multiplied because we are matching strobemers to other strobemers in the grouping step. On the other hand, Sahlin [12] only matched strobemers to a reference genome, so the longer strobemers are tolerated better with similar error rates. We deduced the strobemers of length 30 made it too probable for them to contain an error making it difficult to identify matching strobemers. We had to adjust the parameters so that the total strobemer length was lower. We decided to keep the number of minimizers at three but lower the minimizer length to seven, leading to the strobemers being 21 characters long. With this setup, we observed that the recall increased to a reasonable range and we were satisfied with the chosen values.

Next, the size of the minimizer window $w$ should be chosen so that even if the areas between the windows have insertions and deletions causing the position of the minimizer window to shift from its correct position, it is still likely the correct minimizer can be found within the minimizer window. Thus, the window should contain a good number of minimizer candidates so the majority of them stay in the intended window even if a few insertions or deletions shift the position of the window. On the other hand, we do not want the window to be too big to ensure there is a unique minimizer in the window (which is also affected by the minimizer length $l$). More importantly, if the window is too long, the resulting strobemers are less likely to be unique to their specific position in the genome causing us to group sequences from different areas of the genome making the consensus generation more difficult. We wanted the window length $w$ to be at least $l$ long so there was no character in the window that was shared by all the minimizer candidates. We then chose a few test values $w = 9, 11, 13, 15$ around our initial guess $w = 11$ to see if the guessed value should be adjusted. In these experiments, we used the already chosen $n = 3$ and $l = 7$ parameters, and the initial guess values for the other parameters ($v$ was chosen based on Eq (1)). We searched for 47-mers and 97-mers and measured the precision and recall using the $k$-mers found in the reference genome. The results can be seen in Fig 5.

We observed that the effects of the window size between these values were very small. The precision had a tiny decrease as the window size grew larger. Nevertheless, both precision and recall were good with all the tested values, and we chose to keep our initial guess value $w = 11$ for the rest of the experiments.

Finally, for the strobemer parameters, we are left to set the value for the distance between minimizer windows $v$. We used this parameter to fit the different values of $k$. And as mentioned earlier, the value of $v$ is not fixed and it is used to scale the length of the sequence covered by the strobemer to fit the different $k$-mer lengths.

After this, we were ready to decide the non-strobemer parameters. As previously mentioned, our initial guesses for them were $b = 0.8$, $z = 0.4$, and $a, g, c = 3$. First, we decided to check if the bundling threshold $b = 0.8$ was reasonable. We used the already decided strobemer parameters $n = 3$, $l = 7$, and $w = 11$, and the initial guesses for the other parameters. We tried using values 0.5, 0.6, 0.7, 0.8, and 0.9 for the bundling threshold. The results can be seen in Fig 6. We observed that with the test data set, the difference in precision and recall between these values was minuscule. The experiments did not definitively indicate which value was the best, so we decided to stick with our initial guess $b = 0.8$ for the bundling threshold. We thought this would allow enough flexibility for the sequences to be bundled together even with some errors, without bundling sequences that are obviously not from the same genomic region.

After $b$ was fixed, we moved on to find a value for the relative bundle support threshold $z$. This value determines if the bundle size is big enough for the consensus sequence to be accepted. If the number of sequences in the bundle divided by the number of all sequences in

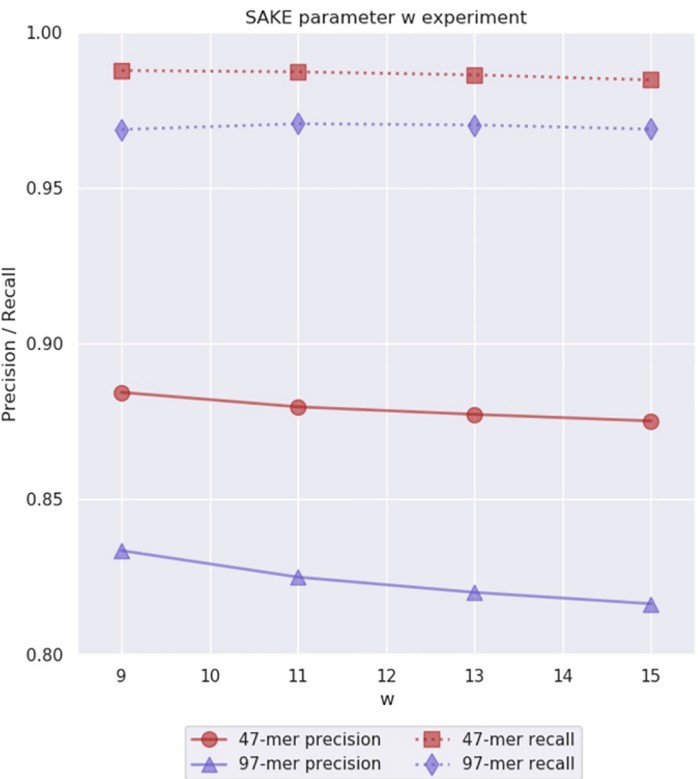

**Fig 5. Parameter *w* experiment results.** This experiment was used to evaluate the initial parameter guess *w* = 11 for the minimizer window length. We performed this experiment on simulated 40x 6% error rate *E. coli* data. Note that the *y*-axis starts from 0.8.

the PO-MSA graph is at least *z*, the consensus sequence is considered to have enough support. Our initial guess for this parameter was 0.4, but we also experimented with values 0.2, 0.3, 0.5, and 0.6. One thing that should be noted about this parameter is that its value determines what is the maximum number of consensus sequences we can produce from a single PO-MSA graph. For example, if one sequence has enough support when *z* = 0.6, then the next consensus sequence can have at most 1−0.6 = 0.4 support because the sequences of the first bundle are removed from the graph. We ran similar tests as before using the chosen strobemer parameters, *b* = 0.8, and initial guesses *a*, *g*, *c* = 3 to search for 47-mers and 97-mers. The result of this experiment can be seen in Fig 7. Again, we saw very little difference between the tested values, which implied that most of the time a single graph produced at most one consensus sequence. Still, we chose to keep the guessed value *z* = 0.4, which allows one graph to produce two consensus sequences while requiring them to have a reasonably large supporting bundle.

Finally, we were left to find what values to choose for parameters *a*, *g*, and *c*. We decided to group these parameters together because they are dependent on each other. For example, if *c* = 5, the characters in a consensus sequence are required to have support from five bundled sequences. If this is the case, it does not make sense to set parameter *g* to be less than five, because if the bundle size was allowed to be four, then no character in the consensus sequence would have enough support. Similarly, it would not make sense to set *a* to be less than five, because the number of strobemer occurrences determines how many sequences appear in the PO-MSA graph, which directly sets a maximum value for the bundle size. Because of these

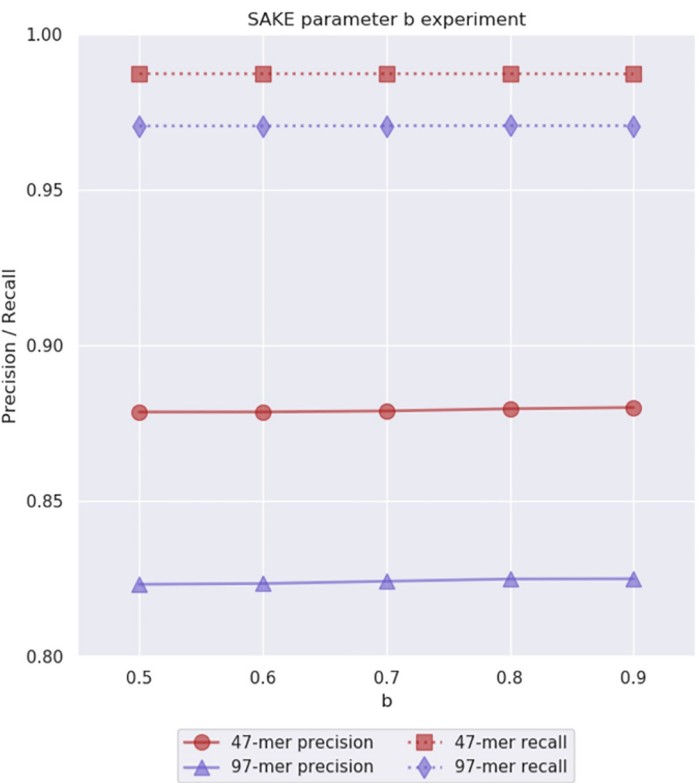

**Fig 6. Parameter *b* experiment results.** This experiment was used to evaluate the initial parameter guess *b* = 0.8 for the bundling threshold. We performed this experiment on simulated 40x 6% error rate *E. coli* data. Note that the *y*-axis starts from 0.8.

parameter dependencies, we decided to treat them as a single parameter when choosing their values.

As our initial guess, we had set all these parameters to be 3. Lower values could not bring better results, so we ran experiments where we set these parameters to be 3, 5, 7, 9, and 11 and used the other already decided parameter values to extract 47-mers and 97-mers. The results of these experiments can be seen in Fig 8. The higher values 7, 9, and 11 did not work well with the data set, so we had to decide between the smaller values. With *a*, *g*, *c* = 3 the recall was better than with *a*, *g*, *c* = 5, while the precision was lower. Looking at these results the best overall value for *a*, *g*, *c* would probably be 4 or 5. Thus, in the Results section, we decided to show experiment results with three different values 3, 4, and 5.

## Results

To evaluate the performance of SAKE, we compared it to LoMeX [8] and DSK [18], which is a *k*-mer counter that uses the exact *k*-mer counting method. SAKE is available on GitHub at https://github.com/Denopia/SAKE.

### Program parameters

SAKE has a few parameters the user can adjust to affect the precision and recall of extracted *k*-mers. One of the most impactful parameters is *a*, the minimum strobemer abundance threshold. This parameter value shows how many sequences a strobemer must have in its group for

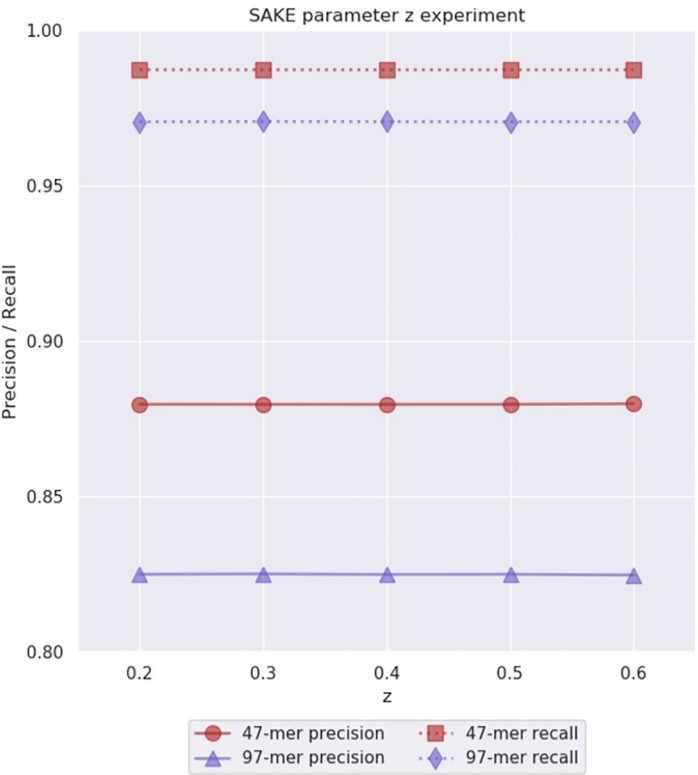

**Fig 7. Parameter *z* experiment results.** This experiment was used to evaluate the initial parameter guess *z* = 0.4 for the relative bundle support threshold. We performed this experiment on simulated 40x 6% error rate *E. coli* data. Note that the *y*-axis starts from 0.8.

the program to produce consensus *k*-mers for that group. A similar parameter *m* is also present in LoMeX, where it determines how many *k*-mers must be in the same group for it to produce consensus *k*-mers. The parameter *a* in SAKE and *m* in LoMeX thus perform a similar function to the *k*-mer abundance threshold *abundance-min* in DSK, where it is the minimum number of times a *k*-mer must occur in the reads for it to be reported. Because we consider abundance parameters *a*, *m*, and *abundance-min* to be analogous to each other, after this point we will refer to all of them with the same letter *a* (which method's abundance we are talking about will be clear from the context).

Since *a* in DSK is the only parameter that should affect the output *k*-mers, we decided to only vary its value and the values of the corresponding parameters in SAKE and LoMeX. Other parameters in LoMeX were set to their default values. The spaced seed parameter determines the length of the extracted *k*-mers. In these experiments, we extracted 47-mers, 57-mers, 67-mers, 77-mers, 87-mers, and 97-mers. The spaced seeds used to find them were:

- $4 - 6 - 4 - 7 - 5 - 7 - 4 - 6 - 4$

- $4 - 9 - 4 - 9 - 5 - 9 - 4 - 9 - 4$

- $4 - 11 - 4 - 12 - 5 - 12 - 4 - 11 - 4$

- $4 - 14 - 4 - 14 - 5 - 14 - 4 - 14 - 4$

- $4 - 16 - 4 - 17 - 5 - 17 - 4 - 16 - 4$

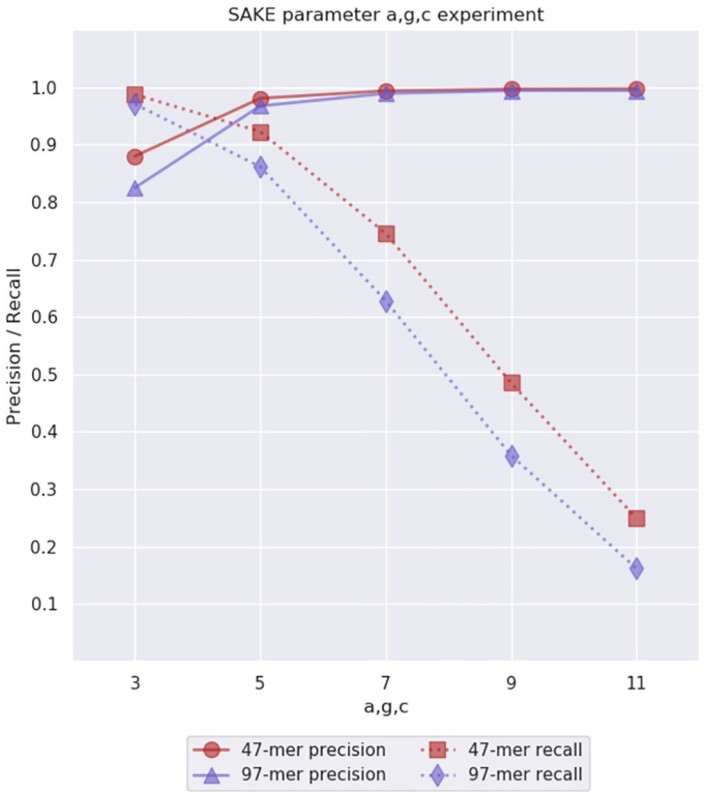

**Fig 8. Parameter *a*, *g*, *c* experiment results.** This experiment was used to evaluate the initial parameter guesses *a*, *g*, *c* = 3 for the minimum strobemer abundance *a*, absolute bundle support threshold *g*, and character support threshold *c*. We performed this experiment on simulated 40x 6% error rate *E. coli* data.

- $4 - 19 - 4 - 19 - 5 - 19 - 4 - 19 - 4$

In this spaced seeds representation the numbers at odd positions show how long the fixed positions streaks are, and the numbers at even positions show how long the "don't care" position streaks between them are. The sum of these numbers is equal to the *k*-mer length. The total number of fixed characters in the spaced seeds was chosen to match the number of characters that appear in the strobemers used by SAKE. There is a similar way to adjust the strobemer parameters in SAKE to match the extracted *k*-mer lengths, which was already discussed in SAKE parameter selection section.

## Experiment setup

We used the three programs to find solid *k*-mers from three data sets: Real *Escherichia coli* reads [22], simulated *Escherichia coli* reads, simulated *Drosophila melanogaster* reads, and real human chromosome 1 reads [23]. Information about the used data sets can be found in Table 2. The use of simulated data allowed us to modulate the error rate so we could experiment and observe how the programs performed with increasing error rates. We also ran an experiment with real reads to gauge the performance with a more realistic error profile. In these experiments, the errors in the simulated reads were uniformly distributed and each error type was equally likely to occur.

**Table 2. Data sets used in the experiments.**

| Organism | Read set and error rate | Reference accession | Reference length | Coverage | Number of reads | Avg. read length |
|---|---|---|---|---|---|---|
| *E. coli* | Real 7.8% error | GCA_000005845.2 | 4,641,652 | 40x | 29,215 | 6,435 |
| *E. coli* | Simulated 4% error | GCA_000005845.2 | 4,641,652 | 40x | 37,140 | 4,999 |
| *E. coli* | Simulated 6% error | GCA_000005845.2 | 4,641,652 | 40x | 37,148 | 4,998 |
| *E. coli* | Simulated 8% error | GCA_000005845.2 | 4,641,652 | 40x | 37,160 | 4,997 |
| *D. melanogaster* | Simulated 6% error | GCA_000001215.4 | 143,726,002 | 40x | 1,125,396 | 4,998 |
| Human Chromosome 1 | Real 4.8% error | CM039011 | 251,946,536 | 69x | 770,676 | 22,383 |

The performance of the programs was analyzed using precision (real reported *k*-mers divided by all reported *k*-mers) and recall (real reported *k*-mers divided by all real *k*-mers). The experiments were run on a computing cluster with eight 15GB cores for a total of 8x15GB = 120GB memory with the exception of human data experiments which were run with eight cores with 300GB memory each. We performed the experiments to extract *k*-mers of varying lengths, between 47bp and 97bp. Other than the length of the *k*-mers, we also varied the abundance threshold *a*. We ran experiments with different *a* values, and for each program, we report results with three *a* values that we considered to give the best tradeoff between precision and recall. For SAKE the best performance was given by *a* values 4, 5, and 6. For LoMeX, *a* values were 3, 4, and 5, and for DSK the three smallest *a* values 2, 3, and 4 gave the best results.

## Precision and recall experiments

We measured how the precision and recall of the extracted *k*-mers by SAKE differs from those of LoMeX and DSK. To examine this, we first experimented with a 40x coverage simulated *E. coli* data set. We used all three programs to extract six different *k*-mers: 47-mers, 57-mers, 67-mers, 77-mers, 87-mers, and 97-mers. We ran these experiments with four different error rates, 4%, 6%, 8%, and 10%. For each program, we did three runs for each error rate with varying abundance parameters. As mentioned earlier, the abundance values for SAKE were 4, 5, and 6, for LoMeX they were 3, 4, and 5, and for DSK they were 2, 3, and 4. The results of these experiments can be seen in Fig 9.

From these experiments, we can see that with the simulated *E. coli* data set, SAKE performs generally better than LoMeX. The recall of SAKE is also always better than that of DSK, while sometimes DSK gets better precision. On the other hand, if we look at the tradeoff between precision and recall, SAKE seems to perform the best, especially with higher values of *k*.

We also wanted to see how these programs performed when *k*-mers were extracted from a longer genome. For this experiment, we simulated a 40x coverage 6% error rate read set from the *D. melanogaster* reference genome. The result of this experiment can be seen in Fig 10.

The larger genome experiments with simulated *D. melanogaster* reads show very similar results to the ones we performed with *E. coli*. Overall, it seems like SAKE is able to give the best tradeoff between precision and recall.

To evaluate how the programs perform on real data, we sampled a 40x read set from a larger set of real Nanopore *E. coli* reads. To estimate the error rate of this read set, we used minimap2 [24, 25] to align the reads to a reference genome and samtools [26] to calculate the error rate. The estimated error rate we got for this data set is 7.8%. The result of this experiment can be seen in Fig 11.

The results between the different methods with real *E. coli* reads are closer to each other compared to what we witnessed with the less realistic simulated reads. Still, SAKE seems to

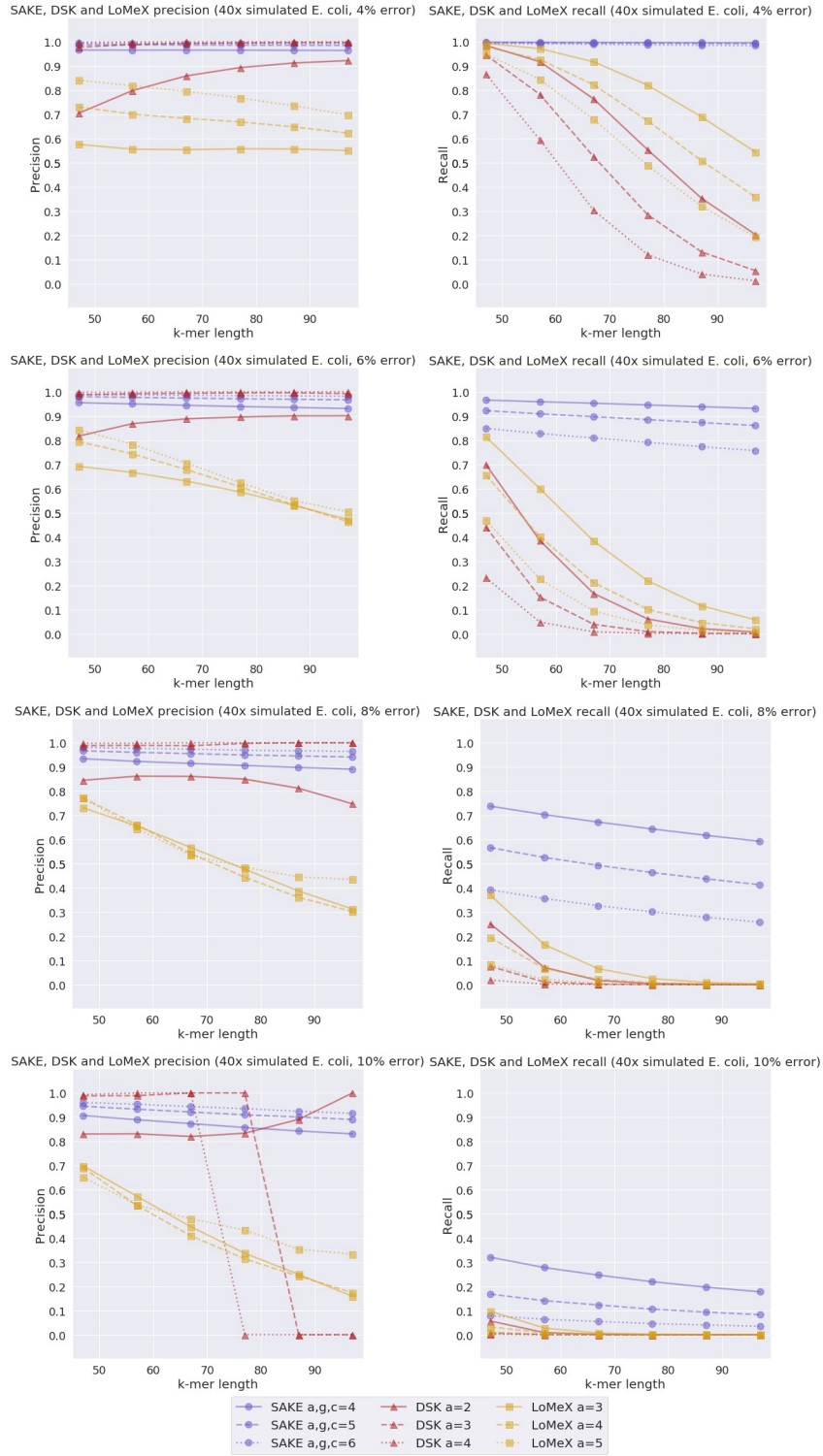

**Fig 9. Simulated *E. coli* experiment results.** SAKE, DSK and LoMeX precision and recall comparison with simulated *E. coli* data with 4%, 6%, 8%, and 10% error rate and 40x coverage.

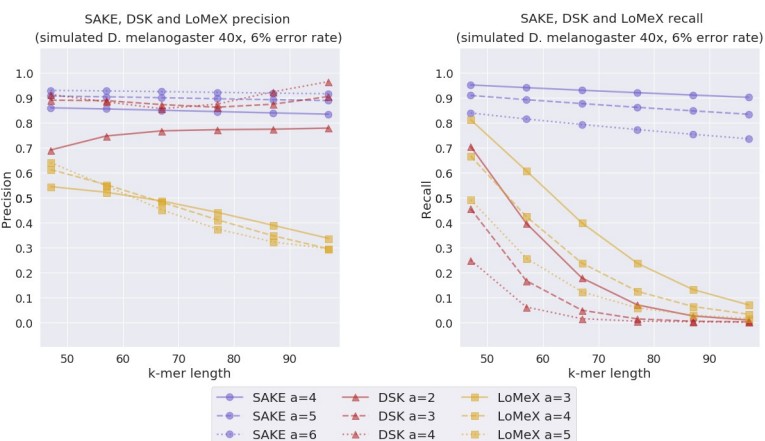

**Fig 10. Simulated *D. melanogaster* experiment results.** SAKE, DSK and LoMeX precision and recall comparison with simulated *D. melanogaster* data with 6% error rate and 40x coverage.

outperform LoMeX, but with smaller values of *k*, SAKE and DSK have similar precision/recall tradeoffs. With bigger values of *k*, DSK admittedly has better precision, but SAKE has a significantly higher recall. Overall, at least with higher *k* values, SAKE seems to produce *k*-mers with the best tradeoff between precision and recall.

## Time and memory usage experiments

Optimizing time and memory usage of SAKE was not our primary goal, but we still recorded them to get a general idea of how SAKE fares against DSK and LoMeX in these aspects. In Table 3, we show the peak memory usages and wall clock times of the programs in twelve experiments. We measured these statistics for different kinds of situations. For all programs, we show two runs for all three data sets, so that the first run is for the shortest *k*-mer, 47-mer, and the second run is for the longest *k*-mer, 97-mer. For SAKE, the output sequences can sometimes be longer than *k*, so they are split into *k*-mers. After this, the *k*-mers are oriented into their canonical form according to standard alphabetical order. Then they are sorted, and

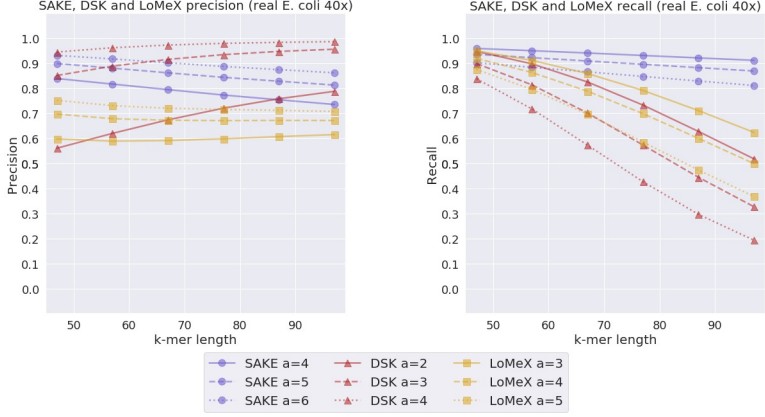

**Fig 11. Real *E. coli* experiment results.** SAKE, DSK and LoMeX precision and recall comparison with real *E. coli* data with estimated error rate 7.8% and 40x coverage.

**Table 3. Time and memory usage comparison between DSK, LoMeX, and SAKE.**

| Program | Data | k | Time [hh:mm:ss] | Memory [kB] |
|---|---|---|---|---|
| DSK | Real *E. coli* | 47 | 00:00:12 | 1,636,244 |
| LoMeX | Real *E. coli* | 47 | 00:07:55 | 3,818,000 |
| SAKE | Real *E. coli* | 47 | 00:15:27 | 5,740,556 |
| DSK | Simulated *E. coli* | 47 | 00:00:11 | 1,549,172 |
| LoMeX | Simulated *E. coli* | 47 | 00:05:12 | 3,809,852 |
| SAKE | Simulated *E. coli* | 47 | 00:12:00 | 5,405,816 |
| DSK | Simulated *D. melanogaster* | 47 | 00:04:26 | 4,763,124 |
| LoMeX | Simulated *D. melanogaster* | 47 | 03:06:35 | 25,435,768 |
| SAKE | Simulated *D. melanogaster* | 47 | 07:11:36 | 15,123,208 |
| DSK | Real *E. coli* | 97 | 00:00:10 | 2,875,696 |
| LoMeX | Real *E. coli* | 97 | 00:07:25 | 3,818,148 |
| SAKE | Real *E. coli* | 97 | 00:21:15 | 5,725,400 |
| DSK | Simulated *E. coli* | 97 | 00:00:17 | 2,768,656 |
| LoMeX | Simulated *E. coli* | 97 | 00:03:27 | 3,811,804 |
| SAKE | Simulated *E. coli* | 97 | 00:15:26 | 5,388,812 |
| DSK | Simulated *D. melanogaster* | 97 | 00:07:03 | 4,705,416 |
| LoMeX | Simulated *D. melanogaster* | 97 | 02:02:15 | 25,435,868 |
| SAKE | Simulated *D. melanogaster* | 97 | 09:40:26 | 15,143,492 |

any unnecessary duplicate *k*-mers are deleted. We have a script for this cleanup step and it is included in the time measurement. We have a similar script to orient the *k*-mers from LoMeX. DSK produces its canonical *k*-mers not according to a standard lexicographic order (instead it is A < C < T < G). We had to orient these *k*-mers too for our precision and recall calculations, but for the time and memory measurements that phase is not included.

From these results, it is clear that in its current form, SAKE is slower and uses more memory than DSK. LoMeX falls between these two programs, except with the *D. melanogaster* data set it uses the most memory. On the other hand, this happened using the default values, and the user can adjust the parameters to decrease memory usage at the expense of running time.

The reason for the high SAKE memory usage is that it keeps all the reads in memory simultaneously. We could improve this by splitting the process into multiple iterations and only keeping in memory the relevant reads or sequences for a smaller number of strobemers. Time-wise, it is unlikely SAKE could be optimized to be faster than DSK since it is not just simply counting the *k*-mers. SAKE spends most of the time building the POA graphs and finding the consensus sequences from them. We used a ready POA implementation called SPOA [19] that we modified slightly to fit SAKE. It is certainly possible that with more optimization we could achieve shorter running times, but the focus of this work was to see how SAKE would perform from the perspective of precision and recall.

## Experiments with corrected reads

Since SAKE performs *k*-mer extraction and also tries to correct errors while doing it, one might question how it fares against DSK if the reads for DSK were corrected first. To find this out, we ran the same real *E. coli* experiment as before, but the reads given to DSK were corrected with a read correction program called CONSENT [15]. SAKE was given the uncorrected original reads. The results from this experiment can be seen in Fig 12.

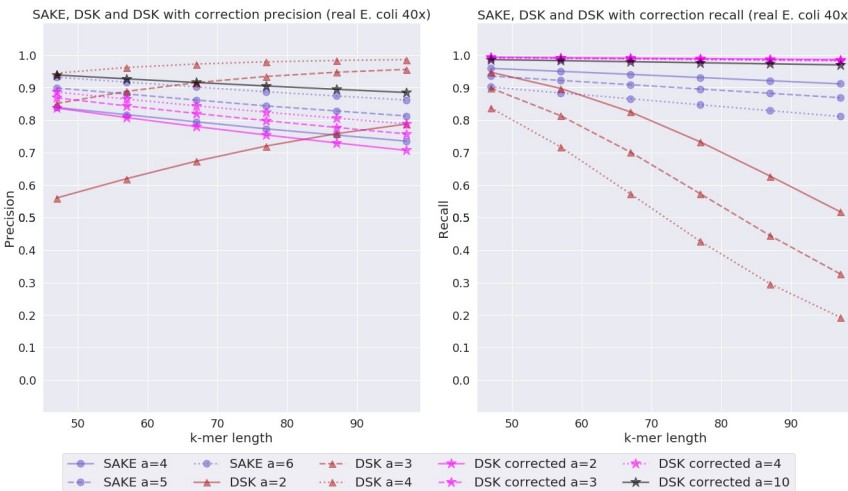

**Fig 12. Corrected real *E. coli* experiment results.** SAKE and DSK precision and recall comparison with real *E. coli* data with estimated error rate 7.8% and 40x coverage. The plot shows the results for DSK with corrected and uncorrected reads, and the results for SAKE only with uncorrected reads.

In addition to this corrected *E. coli* experiment, we ran a similar corrected reads experiment with a human chromosome 1 real read set. Since these experiments took a long time to run, we performed them to only extract 47-mers, 67-mers, and 97-mers. The results from this experiment can be seen in Fig 13. Unfortunately, we were unable to correct the reads with CONSENT so we used the error correction tool NECAT [27] instead.

Fig 12 shows that with the same *a* values as before, DSK is able to find more real *k*-mers on the real *E. coli* data. On the other hand, its precision is generally slightly lower than with the uncorrected reads. With these parameter values, SAKE still seems to have a better precision/recall tradeoff. However, if we raise the abundance threshold of DSK to *a* = 10, then DSK is able to beat SAKE in both precision and recall. This is not too surprising since the error correction process of CONSENT is more thorough compared to the consensus *k*-mer generation of SAKE.

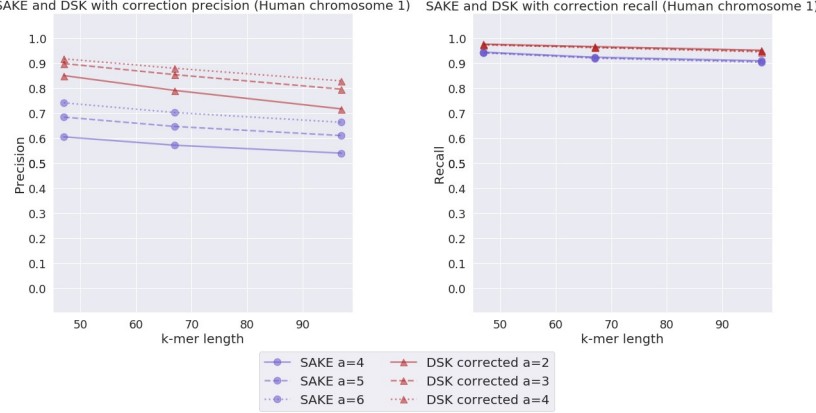

**Fig 13. Corrected real human chromosome 1 experiment results.** SAKE and DSK precision and recall comparison with real Human chromosome 1 data with estimated error rate 4.8%. The reads given to DSK were corrected beforehand with the read correction program NECAT while SAKE still used the original uncorrected reads.

On the other hand, human chromosome 1 experiments show that if the reads are first corrected with NECAT, then DSK gives more accurate results than running SAKE on uncorrected reads. NECAT and DSK combination is also faster than SAKE.

We observed that read correction with CONSENT does not come without a downside. Correcting the reads will take significantly more time compared to only running DSK. For example, correcting the real 40x *E. coli* read set with CONSENT took us 30 minutes and 39 seconds. On the other hand, SAKE took 15 minutes and 27 seconds on this data set. However with NECAT, human data error correction is very efficient. The correction process took 15 hours and 9 minutes. After this, even the longest DSK run was finished within 15 minutes. As comparison, even the fastest SAKE runs took at least 24 hours. However, as stated earlier SAKE is not optimised for speed.

### Extraction after assembly experiments

Related to the experiments with corrected reads, it is interesting to also see how SAKE fares against standard *k*-mer counting when the reads have been assembled beforehand. We decided to use Canu [28] to assemble the reads, and only use real reads to perform the experiment. We set the time limit to 7 days, and unfortunately Canu was unable to assemble the human reads within the time period. However, we managed to run the experiment on the smaller *E. coli* data set. The results of the experiment can be seen in Table 4.

After assembly DSK is very accurate, but the process is slower compared to the SAKE times found in Table 3. SAKE was able to extract 47-mers in under 16 minutes and 97-mers in under 22 minutes. In both cases DSK with Canu too over 36 minutes. The same time usage for DSK with Canu with all values of *k* is explained by the fact that the assembly process is the same in all cases, and the resulting data set (assembled reads) is significantly more compact so DSK is able to extract all k-mers very quickly (total time was around 3 seconds for every value of *k*). The memory peak in every case was achieved during the Canu assembly. One of the drawbacks of assembling the reads before *k*-mer extraction is that we cannot obtain the exact *k*-mer counts of the reads. However, this is a fair comparison against SAKE since it does not report counts either.

Graphical precision/recall comparison against SAKE can be found in Fig 14. From the plot we can observe that assembling the reads before extracting *k*-mers gives higher precision and recall. This is expected, since Canu has its own read correction phase and will produce a good representation of the genome for DSK to extract the *k*-mers from.

### Downstream application experiments

To see how *k*-mers extracted by SAKE might perform in downstream applications, we used BCALM [29] to construct a de Bruijn graph and build unitigs with different extracted *k*-mer

**Table 4. Results of running first Canu on the real *E. coli* data set and then DSK on the assembly produced by Canu.**

| *k* | Precision | Recall | Time [hh:mm:ss] | Memory [kB] |
|-----|-----------|--------|-----------------|-------------|
| 47  | 0.969     | 0.969  | 00:36:44        | 6,393,756   |
| 57  | 0.963     | 0.962  | 00:36:44        | 6,393,756   |
| 67  | 0.956     | 0.956  | 00:36:44        | 6,393,756   |
| 77  | 0.949     | 0.949  | 00:36:44        | 6,393,756   |
| 87  | 0.942     | 0.942  | 00:36:44        | 6,393,756   |
| 97  | 0.936     | 0.936  | 00:36:44        | 6,393,756   |

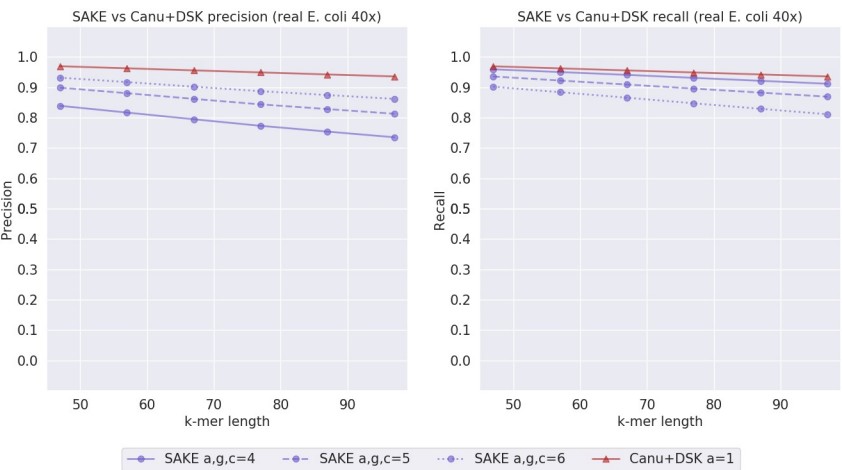

**Fig 14. Extraction after assembly experiments.** SAKE and DSK precision and recall comparison with real *E. coli* data with estimated error rate 7.8% and 40x coverage. SAKE uses the original reads while the DSK was given the assembly results produced by Canu.

sets. This experiment was performed with the *k*-mers extracted from the real *E. coli* data, except only 47-mers, 67-mers, and 97-mers were used. We analyzed the obtained unitigs using QUAST [30] and recorded the NGA50 values and the reference genome coverages in Table 5.

From these results we can observe that longer unitigs can be produced with SAKE*k*-mers compared to plain DSK. However, DSK clearly produces longer unitigs if the reads are corrected beforehand but SAKE still has a better genome coverage.

## Discussion

Experiments with the uncorrected simulated data show that with a uniformly distributed error profile, SAKE gives the best tradeoff between precision and recall compared to the other methods. We observed that SAKE outperforms LoMeX in both metrics. On the other hand, DSK sometimes manages to get higher precision, but in these cases, the recall is very low. The results were similar in both simulated cases: the smaller *E. coli* genome and the bigger *D. melanogaster* genome. Still, the simulated read experiments can only estimate how the *k*-mer extraction could work with real data sets.

The estimated error rate in the experiments with the real uncorrected *E. coli* reads is 7.8%, so one could expect the results will fall somewhere between the 6% and 8% simulated read results. Comparing the results, we can see that the SAKE precision is slightly lower than one could expect from the simulated case, while the recall is considerably better than expected. A similar trend can be seen with LoMeX and DSK: the recall with them is also better than expected. The precision with DSK is slightly worse, while LoMeX precision is better. Some of the surprising behavior between simulated and real reads can be explained by the estimated 7.8% error rate being off, but also the specific error profile of the real Nanopore reads is likely to play a part in the observed differences. Nevertheless, we would still claim that even with this realistic error profile, SAKE provides the most balanced results between precision and recall.

While SAKE looks like it could be the best option based only on these experiments, we should also discuss its shortcomings. First, looking at the resource usage in Table 3, it becomes very clear that at least in its current form SAKE is slower and requires more memory to run

**Table 5. NGA50 and genome fraction of SAKE and DSK with real *E. coli* data.**

| Program | Reads | *k* | *a* | NGA50 | Genome fraction |
|---|---|---|---|---|---|
| DSK | Uncorrected | 47 | 2 | 92 | 95.887 |
| DSK | Uncorrected | 67 | 2 | 136 | 99.359 |
| DSK | Uncorrected | 97 | 2 | 169 | 93.922 |
| DSK | Uncorrected | 47 | 3 | 151 | 96.407 |
| DSK | Uncorrected | 67 | 3 | 171 | 96.560 |
| DSK | Uncorrected | 97 | 3 | 134 | 77.750 |
| DSK | Uncorrected | 47 | 4 | 197 | 94.375 |
| DSK | Uncorrected | 67 | 4 | 144 | 91.325 |
| DSK | Uncorrected | 97 | 4 | 111 | 58.818 |
| DSK | Corrected | 47 | 2 | 490 | 99.026 |
| DSK | Corrected | 67 | 2 | 498 | 99.687 |
| DSK | Corrected | 97 | 2 | 505 | 99.794 |
| DSK | Corrected | 47 | 3 | 598 | 98.991 |
| DSK | Corrected | 67 | 3 | 605 | 99.591 |
| DSK | Corrected | 97 | 3 | 610 | 99.726 |
| DSK | Corrected | 47 | 4 | 694 | 98.972 |
| DSK | Corrected | 67 | 4 | 703 | 99.511 |
| DSK | Corrected | 97 | 4 | 712 | 99.707 |
| DSK | Corrected | 47 | 10 | 1,362 | 98.852 |
| DSK | Corrected | 67 | 10 | 1,390 | 99.240 |
| DSK | Corrected | 97 | 10 | 1,426 | 99.462 |
| SAKE | Uncorrected | 47 | 4 | 257 | 97.686 |
| SAKE | Uncorrected | 67 | 4 | 273 | 99.900 |
| SAKE | Uncorrected | 97 | 4 | 291 | 99.977 |
| SAKE | Uncorrected | 47 | 5 | 285 | 96.890 |
| SAKE | Uncorrected | 67 | 5 | 275 | 99.691 |
| SAKE | Uncorrected | 97 | 5 | 266 | 99.885 |
| SAKE | Uncorrected | 47 | 6 | 258 | 95.343 |
| SAKE | Uncorrected | 67 | 6 | 235 | 99.312 |
| SAKE | Uncorrected | 97 | 6 | 223 | 99.637 |

than DSK, which simply reads the data set and counts how many times each *k*-mer is seen. We believe the method deployed by SAKE can be optimized to lessen the resource requirements, but it was not in the scope of this work.

Second, the results in Figs 12 and 13 show that if the reads are corrected and the abundance threshold is raised to an appropriate value, DSK can beat SAKE in precision and recall. Of course, correcting the reads can take a lot of time depending on the size of the read set and the correction method applied, so taking this approach does not necessarily come without a cost. As mentioned, correcting the real *E. coli* read set took over 30 minutes which is longer than it took SAKE to find even the longest 97-mers. On the other hand, NECAT was able to correct the human chromosome 1 reads in 15 hours, whereas SAKE took 24 hours to extract the *k*-mers. Nevertheless, correcting full reads is unnecessary when we only need the extracted *k*-mers to be correct. The experiments with DSK *k*-mer counting on Canu-assembled data show similar results as the corrected read experiments. Assembling the reads beforehand produces accurate results, but takes significantly longer than just counting the *k*-mers, at least with with the Canu assembler.

Lastly, in its current state, SAKE does not provide *k*-mer counts. In LoMeX, direct counts cannot be given either, but the *k*-mer counts are estimated using the number of grouped *k*-mers and the number of produced consensus *k*-mers. Something similar could be implemented for SAKE, but it is not as straightforward. We could estimate the counts using the number of bundled sequences, but the same *k*-mer can sometimes be produced by two different strobemers and then these two counts for the same *k*-mer would need to be combined. If all the sequences between the strobemers are from different genomic regions, then summing the estimated counts would work. But in the case that there are overlapping sequences between the groups means that then we would overestimate the count. In the end, we did not implement estimating the counts and left it for future work instead.

In summary, SAKE gives the most balanced results out of the three *k*-mer counting/extraction programs regardless of if the reads are simulated or real Nanopore reads if the reads are not corrected. The use of strobemers seems more beneficial compared to using spaced seeds if we want to perform consensus *k*-mer generation. However, with an optimized error correction tool at your disposal, the best option still seems to be to correct the reads and use a standard *k*-mer counting method.

## Conclusion

In this paper, we proposed SAKE, a strobemer-assisted method for *k*-mer extraction. The aim of SAKE is to enable the extraction of long *k*-mers from read sets with error rates where the exact *k*-mer counting method fails. SAKE was also designed to outperform LoMeX [8] which is a *k*-mer extraction program that can find *k*-mers in the presence of substitution errors. Furthermore, SAKE was intended to also work in the presence of insertion and deletion errors. Our experiments with data containing insertions and deletions show that SAKE performs well against LoMeX. SAKE can also give a better balance between precision and recall compared to DSK [18], a program for exact *k*-mer counting. In fact, DSK becomes unreliable/unusable more quickly compared to SAKE as the *k*-mer length increases.

Only when reads are corrected before giving them to DSK, can it find *k*-mers with higher precision and recall. This is not too surprising, since the error correction part of SAKE is not as thorough as that of a dedicated read correction tool. While correcting full reads before doing exact *k*-mer counting can give better results than SAKE, it can be time consuming. However, at the moment, the performance of SAKE is not optimized. Thus, correcting the reads with an optimized error correction tool before counting the *k*-mers in the standard way is still the better option. However, we believe that instead of correcting full reads, the *k*-mer correcting approach of SAKE could be a more lightweight option. Our intuition is that with a more optimized implementation, SAKE *k*-mer extraction can be a more efficient alternative to correcting the reads and counting *k*-mers afterward.

## Author Contributions

**Conceptualization:** Miika Leinonen, Leena Salmela.

**Funding acquisition:** Leena Salmela.

**Investigation:** Miika Leinonen.

**Methodology:** Miika Leinonen, Leena Salmela.

**Software:** Miika Leinonen, Leena Salmela.

**Supervision:** Leena Salmela.

**Visualization:** Miika Leinonen.

**Writing – original draft:** Miika Leinonen, Leena Salmela.

**Writing – review & editing:** Miika Leinonen, Leena Salmela.

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
