## [Decision Letter · Decision Letter 0]

26 Sep 2023

PONE-D-23-17928SAKE: Strobemer-assisted k-mer extractionPLOS ONE

Dear Dr. Leinonen,

Thank you for submitting your manuscript to PLOS ONE. After careful consideration, we feel that it has merit but does not fully meet PLOS ONE’s publication criteria as it currently stands. Therefore, we invite you to submit a revised version of the manuscript that addresses the points raised during the review process.

We look forward to receiving your revised manuscript.

Kind regards,

Alfonso Esposito, PhD

Academic Editor

PLOS ONE

Reviewers' comments:

Reviewer's Responses to Questions

**Comments to the Author**

1. Is the manuscript technically sound, and do the data support the conclusions?

Reviewer #1: Yes

Reviewer #2: Yes

2. Has the statistical analysis been performed appropriately and rigorously? 

Reviewer #1: Yes

Reviewer #2: N/A

3. Have the authors made all data underlying the findings in their manuscript fully available?

Reviewer #1: Yes

Reviewer #2: Yes

4. Is the manuscript presented in an intelligible fashion and written in standard English?

Reviewer #1: Yes

Reviewer #2: Yes

5. Review Comments to the Author

Reviewer #1: It is a well-written and insightful piece of work that presents a valuable contribution to the field of bioinformatics. Your tool's approach to k-mer extraction is both innovative and promising, offering great potential for various applications.

Reviewer #2: Leinonen and Salmela's manuscript "SAKE: Strobemer-assisted k-mer extraction" tackles the problem of determining the presence of **long** k-mers in the ~100-mer regime from a sequenced but unaligned error-prone long-reads (e.g. Oxford Nanopore data). They propose a modification to the Strobemer scheme of Sahlin and company, paired with a partial-order alignment, to allow for finding k-mers even in the presence of indel errors. Overall, the paper is well-motivated and a pleasure to read. Additionally, all code and data appear to be available.

Important note: although SAKE is compared to DSK and LoMeX, SAKE is most comparable to LoMeX (by the same pair of authors), which is also a k-mer extractor, but **not** a k-mer counter. On the other hand, the substantially more-used DSK counts k-mers, rather than simply establishing presence, providing strictly more information.

In terms of accuracy for long k-mers in the presence of indel error, SAKE is a clear winner over both DSK and LoMeX. DSK does not attempt to deal with errors, while LoMeX only handles substitution error, and this shows in the benchmarking. Unfortunately, this comes at a substantial cost in terms of time, as SAKE is also much slower. Indeed, the authors also ran a much-needed benchmark on corrected reads, where the advantage of SAKE more or less disappeared. From a practical point of view, at the present time, from a computational perspective, it's hard to make a clear decision of SAKE over correction+DSK, though I commend the authors on including that benchmark.

I personally would've liked to see a benchmark comparing SAKE with assembly or mapping followed by k-mer extraction on the assemblies. One of the problems with framing the task as k-mer extraction instead of k-mer counting is that the results are a strict subset not just of the k-mer counting problem, but also the assembly problem, and SAKE thus has to compete with very well-optimized assemblers, for that is a major bioinformatics task. Looking at the runtimes, I think SAKE likely has a substantial edge on assembly, but not necessarily on mapping, and it would be useful to the end-reader to see that comparison. Personally, given the maturity of assemblers and mappers, I think that in most settings where I'd want long k-mer extraction, I'd probably just go with those tools instead, so it'd be helpful for the authors to comment specifically on more cases where k-mer extraction fulfills a bioinformatics need.

Overall, I think what makes SAKE most interesting is in its adaptation of strobemers to compensate for indels in the reverse-complement setting of k-mer extraction. This technical contribution I could see taking on a life of its own outside of SAKE, and is I think sufficient even alone to merit a place in the published scientific record (as PLOS ONE describes in its reviewer guidelines). As such, SAKE as a whole certainly is a nice little tool which I could see being usefully applied in certain niche settings.

6. PLOS authors have the option to publish the peer review history of their article (what does this mean?). If published, this will include your full peer review and any attached files.

Reviewer #1: No

Reviewer #2: No

---

## [Author Response · Author response to Decision Letter 0]

20 Oct 2023

We have responded to the editor's and reviewers' comments in the submitted "Cover Letter" and "Response to Reviewers", respectively. We hope our responses can be accessed by the intended recipients without rewriting them separately here.

---

## [Editor Report · Decision Letter 1]

31 Oct 2023

SAKE: Strobemer-assisted k-mer extraction

PONE-D-23-17928R1

Dear Dr. Leinonen,

We’re pleased to inform you that your manuscript has been judged scientifically suitable for publication and will be formally accepted for publication once it meets all outstanding technical requirements.

Kind regards,

Alfonso Esposito, PhD

Academic Editor

PLOS ONE
---

## [Editor Report · Acceptance letter]

16 Nov 2023

PONE-D-23-17928R1 

SAKE: Strobemer-assisted k-mer extraction 

Dear Dr. Leinonen:

I'm pleased to inform you that your manuscript has been deemed suitable for publication in PLOS ONE. Congratulations! Your manuscript is now with our production department. 

Kind regards, 

on behalf of

Prof. Alfonso Esposito 

Academic Editor

PLOS ONE